# Unveiling the Role of Histone Methyltransferases in Psoriasis Pathogenesis: Insights from Transcriptomic Analysis

**DOI:** 10.3390/ijms26136329

**Published:** 2025-06-30

**Authors:** Dóra Romhányi, Ágnes Bessenyei, Kornélia Szabó, Lajos Kemény, Rolland Gyulai, Gergely Groma

**Affiliations:** 1Department of Dermatology and Allergology, University of Szeged, H-6720 Szeged, Hungary; romhanyidora9411@gmail.com (D.R.); agnesbessenyei99@gmail.com (Á.B.); szabo.kornelia@med.u-szeged.hu (K.S.); kemeny.lajos@med.u-szeged.hu (L.K.); gyulai.rolland@med.u-szeged.hu (R.G.); 2Hungarian Centre of Excellence for Molecular Medicine-University of Szeged Skin Research Group (HCEMM-USZ Skin Research Group), H-6720 Szeged, Hungary; 3HUN-REN-SZTE Dermatological Research Group, H-6720 Szeged, Hungary

**Keywords:** epigenetics, histone methyltransferases, immune responses, proliferation, psoriasis

## Abstract

Psoriasis involves complex epigenetic alterations, but detailed studies on histone methyltransferases and their role in disease progression are limited. We conducted a comprehensive analysis of nearly 300 transcriptomes, focusing mainly on differential expression of protein isoform-coding transcripts within the SET domain family of histone methyltransferases. Consistent with previous findings, EZH2 transcripts showed increased expression in lesional skin, indicating altered H3K27 methylation that may enhance gene silencing, promoting keratinocyte proliferation and inflammatory responses. In the SET2 family, ASH1L exhibited reversed expression patterns between non-lesional and lesional skin, while NSD1 and NSD2 were upregulated, and SETD2 downregulated in lesions, suggesting disrupted H3K36 methylation that may affect immune responses and keratinocyte proliferation. Among H3K9 methyltransferases, SUV39 members, SUV39H2 was upregulated in lesions, whereas EHMT1 transcripts increased in non-lesional skin, and SETDB2 decreased in lesions. Additionally, PRDM family members such as PRDM2, MECOM (PRDM3), PRDM6, and PRDM8 showed altered expression in lesional skin. The H4K20 methylating SUV4-20 subfamily member, a SUV420H1 transcript, and SETD8 belonging to the other SET domain-containing family of methyltransferases were significantly increased in non-lesional skin and in lesions, respectively. Overall, aberrant expression and isoform variability of histone methyltransferases likely contribute to psoriasis pathogenesis by dysregulating proliferation, differentiation, and immune responses.

## 1. Introduction

Psoriasis, affecting 2–3% of the population [1], is an immune-mediated skin disease characterized by inflammation and an exaggerated response to stressors, leading to abnormal keratinocyte proliferation and immune cell infiltration and response [2]. In psoriasis, numerous molecular and cellular alternations occur even in the symptom-free non-lesional skin, some of which stabilize the non-lesional skin state while others set the stage for lesion development [3]. Alterations of non-lesional psoriatic skin are not limited to the cells but also affect the extracellular matrix, including modified splicing [4], processing [5], and degradation [6] of extracellular molecules, some of which are known to have an impact on cell proliferation [7,8]. Considering these complex non-lesional skin abnormalities, epigenetic dysregulations were proposed to play a role [9], potentially affecting keratinocyte proliferation, differentiation, and immune responses.

Among epigenetic regulatory processes, histone methylation plays a crucial role in the regulation of proliferation. Histone methylation patterns of proliferating cells undergo specific modulation during different phases of the cell cycle [10]. Mono- and dimethylated ‘Lys-9’ of histone H3 (H3K9me1/2) remain unchanged throughout the cell cycle, while H3K9me3 shows a pronounced peak during the late G2 to mitosis (M) transition [11]. Monomethylated ‘Lys-20’ of histone H4 (H4K20me1) peaks during G2 to M transition but quickly converts to dimethylated ‘Lys-20’ of histone H4 (H4K20me2), which remains consistently high throughout the cell cycle, while H4K20me3 shows a slight increase in early G1 phase [12]. Disturbances in these regulatory processes can lead to an overturning of the cell division rate. During epidermal development, trimethylated ‘Lys-27’ of histone H3 (H3K27me3) and trimethylated ‘Lys-20’ of histone H4 (H4K20me3) methylation levels transition from low in basal cells to high in suprabasal cells, suggested to be important for the proper switch from proliferation to differentiation [13]. Indeed, abnormal keratinocyte differentiation in psoriasis has been shown to be accompanied by altered histone methylation patterns [14]. The higher levels of H3K27me3 described in psoriatic skin [15] suggest aberrant epigenetic regulation, which is likely to contribute to hyperproliferation and abnormal differentiation of keratinocytes.

Dynamic transitions of histone methylation patterns are also important regulators of both adaptive and innate immune responses. Dysregulation of immune cell functions plays a crucial role in the development and maintenance of psoriatic lesions, among which T cell-mediated immune responses are believed to play a central role [16]. The interplay of bivalent chromatin marks, such as trimethylated histone H3 (H3K4me3 and H3K27me3), appears to regulate important T cell functions, including differentiation, effector/memory T cell formation, T cell exhaustion processes, and differentiation of helper T cells into different subsets [17]. Dynamic regulation of H3K27me3 is also observed in natural killer cells and macrophages, and these processes balance pro-inflammatory and immunomodulatory activities to maintain immune homeostasis [18,19].

Despite the high impact of histone methylation-related alterations in psoriasis, a complete overview of this process in the context of the disease is still missing. Therefore, to fulfill this gap, we aimed to provide an overview of histone methyltransferases and their expression in psoriasis by analyzing a literature-based [20,21,22] psoriasis transcriptome database [23] of nearly 300 individuals. Finally, we also analyzed histone methyltransferases identified with altered expression for their possible associations with key disease-related processes, including cell proliferation/differentiation and immune regulation.

## 2. Results

To identify histone methyltransferases with altered expression in psoriasis, we analyzed the transcriptional profiles of all lysine and/or arginine methyltransferases belonging to the two known major families of SET domain (Figure 1a) and 7β-strand methyltransferases (Figure 1b). These molecules function in large complexes [24,25]; therefore, we have included components of these complexes in our analysis (Figure 1c). The resulting methyltransferases and methyltransferase complex members identified with differentially expressed transcripts in psoriasis are shown in Figure 1d (and Appendix A).

### 2.1. SET Domain-Containing Histone Lysine Methyltransferases with Altered Expression in Psoriasis

The most characteristic histone lysine methyltransferases belong to the SET domain family (Figure 1a). Among the family members, we identified EZH1/2, KMT5A, MECOM, NSD1-3, SETD2, SETDB2, SUV39H2, PRDM2, and PRDM8 with differential expression in lesional skin and EHMT1/2 in non-lesional skin (Figure 1d and Figure 2, Table 1 and Appendix A). Meanwhile, ASH1L and SUV420H1 displayed abnormal expressions in both non-lesional and lesional skin (Figure 1d and Figure 2, Table 1 and Appendix A). Our analysis also unveiled transcriptional disparities of family members with uncertain histone methyltransferase activity in lesional skin, including SETD3, SETD4, and SETD6, whereas SETD5 exhibited differences in both non-lesional and lesional skin compared to healthy controls (Figure 1d and Figure 2, and Appendix A).

### 2.2. Histone Lysine Methyltransferase Complex Members Affected by Altered Expression in Psoriasis

Among the members of the histone methyltransferase COMPASS complex (Figure 1c and Figure 2), CXXC1 shows changes in expression in non-lesional skin (Figure 1d and Figure 2, Table 2 and Appendix A). The expression of ASH2L, which functions as a member of both the COMPASS and COMPASS-like complexes (Figure 1c and Figure 2), was found to be altered in both lesional and non-lesional skin (Figure 1d and Figure 2, Table 2 and Appendix A), while KDM6A and MEN1, members of a COMPASS-like complex (Figure 1c and Figure 2), displays alteration only in lesional skin (Figure 1d, Table 2 and Appendix A).

EZH1/2 are the catalytic methyltransferase subunits of the PRC2 complex [64,65] (Figure 1c and Figure 2), of which the non-catalytic members EED and RBBP4 show abnormal expression in non-lesional and lesional skin (Figure 1d, Table 2 and Appendix A). The PRC2 complex has two modules, the PRC2.1 and the PRC2.2 subcomplexes [66] (Figure 1c). In psoriatic skin, we have observed expression changes of the PRC2.1 components PHF1 and MTF2 in non-lesional samples, and PHF19 and EPOP in skin lesions (Figure 1d and Figure 2, Table 2 and Appendix A). In contrast, members of the PRC2.2 subcomplex were found to be unaffected in both psoriatic non-lesional and lesional skin.

### 2.3. Alterations in the Expression of Seven-β-Strand Lysine Methyltransferases in Psoriasis

While seven-β-strand methyltransferases are predominantly recognized as non-histone-specific enzymes, several members have been identified as histone methyltransferases [67,68] (Figure 1b and Table 3). Among the members known for their histone methyltransferase activity, only DOT1L was identified with altered expression in lesional (but not in non-lesional) psoriatic skin, compared to the healthy controls (Figure 1d, Table 3 and Appendix A).

We observed abnormal transcriptional expression of the known non-histone-modifying lysine methyltransferases EEF2KMT, METTL12, and VCPKMT in psoriatic lesional skin, while METTL13 showed alterations in non-lesional skin (Figure 1d, Table 3 and Appendix A). In addition, both non-lesional and lesional skin exhibited abnormalities in METTL21A expression (Figure 1d, Table 3 and Appendix A). Table 3 summarizes the differentially expressed lysine seven-β-strand methyltransferases, including their targets, types, and modification sites.

### 2.4. Modifications in the Expression of Seven-β-Strand Arginine Methyltransferases in Psoriasis

Protein arginine methyltransferases belonging to the seven-β-strand group possess histone- and non-histone-specific methyltransferase activity (Figure 1b). Among the histone-arginine methyltransferases (PRMTs), four members (CARM1 also known as PRMT4 and PRMT1/2/7) exhibited altered expression only in lesional skin (Figure 1d and Figure 3, Table 4 and Appendix A), whereas PRMT5 displayed expression changes in both lesional and non-lesional skin (Figure 1d and Figure 3, Table 4 and Appendix A). In addition, the METTL23 arginine methyltransferase, which shares only distant homology with PRMTs, was identified with expression changes in lesional skin (Figure 1d and Appendix A). Differentially expressed PRMTs, along with their target histones, modification types, and sites, are summarized in Table 4 and Figure 3.

### 2.5. Diversity of Methyltransferase Transcript Variants and Encoded Isoforms in Psoriasis

Alternative splicing generates a diverse pool of transcript variants, including non-protein-coding transcripts and those that encode protein isoforms with altered or novel functionalities compared to the canonical form [70]. These alternative isoforms can exhibit distinct or even antagonistic biological roles relative to their canonical counterparts [71,72]. To investigate this phenomenon in psoriasis, we analyzed the composition of potential protein isoforms coded by differentially expressed transcripts (DETs) in both non-lesional and lesional skin samples, compared to healthy controls.

Within the EZ family, we identified a non-protein-coding transcript of EZH1 containing a retained intron, which was significantly decreased in lesional skin (Table 5). Conversely, four DETs of EZH2, with increased expression in lesional skin, encoded the canonical isoform (Q15910-1) along with three additional transcript variant-coded isoforms (Q15910-2-4) that retained all essential functional domains; however, differences from the canonical isoform affect their DNMT binding site, suggesting potential functional diversification (Table 5 and Appendix A and Figure 4A). In case of Q15910-4 isoform, a glycosylation and two phosphorylation post-translational modification sites are missing that may affect its localization or catalytic activity.

In the SET2 family, we identified DETs of ASH1L, NSD1, NSD2, and SETD2 in our database. The same ASH1L transcript variant was upregulated in non-lesional skin and downregulated in lesional tissue, coding for a 154-amino-acid-long isoform (H0YI82) that partially overlaps with the Bromo domain and fully with the PHD finger domain of the canonical protein (Table 5 and Appendix A, Figure 4B). The NSD1 DET observed in lesional skin encodes a truncated isoform (A0A8I5QJP2) that lacks the first 291 amino acids. For NSD2, increased expression was observed for both the canonical protein isoform (O96028-1)-coding transcript and for a non-protein-coding DET in lesional samples. Additionally, a non-protein-coding processed transcript of NSD3 was found in lesions with decreased expression (Table 5). The SETD2 DET found in lesions encodes an isoform (H7BXT4) that overlaps with the canonical protein sequence from amino acids 130 to 1487 but lacks any known functional domains, suggesting it may be an inactive and, potentially, a regulatory isoform (Figure 4B and Appendix A).

Analyzing the SUV39 family, we identified DETs of EHMT1, EHMT2, SETDB2, and SUV39H2. Specifically, EHMT1 transcripts showed increased expression of both a non-protein-coding and a protein-coding transcript in non-lesional skin. The latter variant encodes a truncated isoform (A0A1W2PPZ7) that lacks nearly all functional domains except for a Cys-rich region overlapping with the canonical Ehmt1 isoform (Table 5 and Appendix A, Figure 4C). A non-protein-coding processed transcript of EHMT2 was found to be upregulated in non-lesional skin samples. A SETDB2-derived transcript exhibited decreased expression in lesional samples, coding for an 11-amino-acid shorter isoform (Q96T68-2) (Table 5 and Appendix A, Figure 4C). SUV39H2 presented two isoform-coding transcripts with elevated expression in lesional samples, one encoding the canonical protein (Q9H5I1-1) and the other a shorter isoform (H0Y306) containing a SET domain (Table 5 and Appendix A, Figure 4C).

Within the SUV4-20 family, the canonical SUV420H1 (KMT5B) isoform-coding transcript was upregulated in lesional skin, while a transcript encoding a C-terminal truncated isoform containing a SET domain (Q4FZB7-2) was increased in non-lesional tissue (Table 5 and Figure 4D).

Among other SET domain-containing histone methyltransferases, only a SETD8 (KMT5A) transcript variant showed elevated expression in lesional skin, encoding a shorter isoform (C9JKQ0) with a reduced-sized SET domain (Table 5 and Appendix A, Figure 4E).

Regarding the PRDM family members with methyltransferase activity, we identified DETs of PRDM2, MECOM (PRDM3), and PRDM8 exclusively in lesional skin (Table 5). Notably, the PRDM2 transcript variant upregulated in lesions encodes a 44-amino-acid-sized micropeptide (H09J3) that overlaps the N-terminal region of the canonical isoform by 35 amino acids (10–44). In lesions, MECOM’s overexpressed transcript variant encodes an isoform (Q03112-1) that lacks the PR domain at the N-terminus (Table 5 and Appendix A, Figure 4F). Conversely, transcripts coding for the canonical isoforms of PRDM8 (Q9NQV8-1) showed decreased expression in lesional tissue.

Among PRDM family members lacking known methyltransferase activity, two PRDM1 transcripts were increased in lesions, encoding the canonical isoform (O75626-1) and a shorter isoform (O75626-2) missing 36 amino acids from the N-terminus. The canonical isoform of PRDM6 (Q9NQX0-3) displayed reduced expression in lesional samples. Additionally, two transcripts of PRDM10, both coding for shorter isoforms that retain all functional domains, exhibited altered expression in lesions. In non-lesional skin, an increase of a ZFPM2 transcript was detected, coding for a shorter isoform of Zfpm2 (E7ET52) with a partial PR domain and an additional zinc finger domain not present in the canonical protein (Table 5 and Appendix A, Figure 4G).

## 3. Discussion

Several studies have investigated epigenetic modifications and aberrant methylation patterns in psoriasis [73,74,75]. However, based on the available knowledge to date, a comprehensive investigation of the histone methyltransferases responsible for shaping histone methylation patterns has not yet been conducted, and only a limited amount of information on how they may regulate proliferation and the immune system dysfunction in psoriasis. Therefore, we performed a detailed analysis of a literature-based psoriasis transcriptome database of nearly 300 individuals to identify differential expression of histone methyltransferases. To provide a complete overview, we discuss the observed expressional alterations and their potential implications in psoriasis of each methyltransferase family (Table 6).

### 3.1. Histone Methyltransferase-Related Alterations in Psoriasis

#### 3.1.1. SET Domain Methyltransferases

The SET domain MTase family is recognized to encompass all known lysine methyltransferases involved in the methylation of flexible histone tails [112,113]. Within the SET domain family, several subfamilies are distinguished by structural differences, including EZ, SET1, SET2, SMYD, SUV39, SUV420, and RIZ (PRDM). Some members are not classified into these subfamilies, such as SET7/9 and SET8 [113]. We refer to these as “other SET domain-containing histone methyltransferases” in our discussion.

##### EZ Subfamily of Methyltransferases

EZH1/2 in the EZ subfamily of methyltransferases is initially inactive [24,26,114] but activates within the PRC2 complex to methylate H3K27 [26], crucial for PRC2-mediated gene silencing to maintain stem cell functions [115,116]. Our analysis showed differential expression of EED, EZH1/2, and RBBP4 in the PRC2 complex, likely to affect stem cell self-renewal [117] and possibly contributing to keratinocyte hyperproliferation in psoriasis [118]. EED modulates T cell immune responses, impacting thymocyte maturation and CD4+ T cells [119,120]. EZH2 was previously shown to relate to keratinocyte proliferation and inflammatory responses in psoriasis [15,76], and it may also affect CD4+ and CD8+ T cell differentiation [121,122] and epidermal stratification [123,124], potentially contributing to psoriatic hyperkeratosis [125,126]. We found increased expression of the canonical and three functional EZH2 isoform-coding transcripts in lesional skin. Sequential differences at the DNMT binding sites of the three non-canonical isoforms may suggest potential functional diversification and may influence DNA methylation. Two DET-coded isoforms (Q15910-2 and Q15910-3) were previously characterized as EZH2α and β, which participate in similar biological processes, but form separate repressive complexes capable of cell-specific gene regulation [127].

PRC2 has two subcomplexes: PRC2.1 and PRC2.2 [128]. In our study, PRC2.1 subcomplex (EPOP, MTF2, PHF1, PHF19) showed transcriptional abnormalities. EPOP influences the chromatin environment and gene expression during the cell cycle G1 phase and aids in the induction of cell differentiation [129]. In addition, MTF2 and PHF19 promote, whereas PHF1 suppresses, cell proliferation and may impact keratinocyte proliferation [130].

##### SET1 Subfamily of Methyltransferases

The SET1 family influences euchromatin-like H3K4 methylation associated with transcriptional activation [131]. SET1 proteins, with low intrinsic activity, assemble into COMPASS and COMPASS-like complexes for enhanced catalytic function [25]. COMPASS di- and trimethylates H3K4 globally [132], while COMPASS-like complexes mono- and dimethylate development-specific genes [28]. These complexes include SET1A/B and four COMPASS-like multiprotein complexes: MLL1-4 [28,133]. Although we found no differences in catalytic subunit expressions, other components of the complex, including ASH2L, CXXC1, KDM6A, and MEN1, exhibited differential expression.

ASH2L regulates pluripotency and cellular reprogramming genes [134]. CXXC1 is crucial for thymocyte development [135], balancing Th1/Th2 [136] and Th17/Treg dynamics [137] relevant to psoriasis [90,111,138,139,140]. KDM6A, a member of the COMPASS-like complex, possesses demethylase activity that counteracts the PRC2 complex by demethylating H3K27me3 and facilitating H3K4me, thereby enhancing IFN responses and tumor-suppressive gene expression [141]. Additionally, KDM6A is vital for lineage-specific differentiation and hematopoietic balance [142,143,144] and contributes to age-related keratinocyte proliferation/differentiation imbalances [88,98,99] that may be important in the late-onset of the disease [145]. Its role in H3K27me3 demethylation affects T cell development [146] and migration [147] and may influence psoriasis pathology through IFN-γ-induced chemokines and T cell recruitment [148,149].

##### SET2 Subfamily of Methyltransferases

The SET2 subfamily, including ASH1L, NSD1-3, and SETD2, orchestrates H3K36 methylation [150], critical for transcriptional activation by SETD2 and H3K36me3 [151]. Our analysis revealed altered expression of ASH1L, NSD1-3, and SETD2. ASH1L maintains epidermal homeostasis, regulates keratinocyte proliferation and differentiation activity [77], and suppresses TLR-mediated inflammatory responses [78]. The ASH1L transcript variant was upregulated in non-lesional skin and downregulated in lesions. Although this ASH1L transcript encodes a non-functional isoform, its PHD finger domain may interfere with the recognition of histones and chromatin modifications, in a contrary manner in non-lesional and lesional skin. The NSD1 DET overexpressed in lesional skin encodes a truncated but functional isoform that may influence chemokine expression and immune cell infiltration via the NF-κB pathway [81,152,153]. Reduced expression of Wnt10b has been detected in psoriatic skin compared to healthy skin [83], and plays a pivotal role in cell proliferation and migration through the NSD1/H3/Wnt10b pathway [82]. In the case of NSD2, we detected the increased expression of the canonical protein isoform-coding transcript in lesions. NSD2 also modulates cell proliferation through the Wnt signaling pathway by regulating cyclin D1 transcription [84], known to be increased in psoriatic lesions [85]. A decreased expression of the SETD2 transcript codes for a non-functional isoform that may interfere with the interaction of the functional isoform in lesions. SETD2 deficiency was previously shown to trigger enhanced keratinocyte proliferation [87], and it influences Th17/Treg balance [86]. Therefore, dysregulated SETD2 may contribute to psoriasis symptoms and immune dysregulation [89,90,139].

##### SMYD Subfamily of Methyltransferases

The SMYD subfamily, comprising SET and MYND domain-containing proteins, plays a dual role, controlling both transcriptional activation and repression of genes [154]. Based on our analysis, none of the members (SMYD1-5) show significant alterations in either non-lesional or involved skin.

##### SUV39 Subfamily of Methyltransferases

The SUV39 subfamily deposits methyl groups onto histone H3 at lysine 9, forming H3K9me2 and H3K9me3 marks [155]. These marks are linked to transcriptional repression and heterochromatin formation [156,157] and are inherited following cell division [158].

We found altered expression of SUV39 subfamily members, including EHMT1, SETDB2, and SUV39H2, in our analysis. H3K9 methylation regulates IL-23 expression through the TNF/N-WASP/EHMT1-2 pathway [73].

An increased expression of EHMT1 transcript variant was observed in non-lesional skin, coding for a shorter isoform containing a Cys-rich region. Cys-rich regions of methyltransferases are known to play a role in maintaining their activity and specificity. Therefore, the shorter isoform may interfere with these properties of EHMT1. This might be relevant in psoriasis since EHMT1 negatively regulates gene induction pathways mediated by NF-κB and type I interferon [91], and is involved in Treg cell differentiation [92]. EHMT1 via CDKN1A modulation regulates the cell cycle [159].

SETDB2 is involved in proliferation-associated chromosome condensation and segregation [43] and inhibits inflammatory cytokine gene transcription via NF-κB [95]. Therefore, the decreased expression of a shorter but likely functional isoform-coding SETDB2 transcript found with reduced expression in lesions may influence these processes. Meanwhile, SUV39H2, found to be elevated in lesions, may modulate the suppression of key genes for epidermal differentiation [97]. Therefore, SUV39 subfamily-related alterations will likely impact immune responses and keratinocyte proliferation and differentiation in psoriasis [160].

##### SUV4-20 Subfamily Methyltransferases

The SUV4-20 subfamily members, SUV420H1 and -H2, serve as methyltransferases primarily responsible for the di- and trimethylation of histone H4K20 for heterochromatin formation and gene silencing [47,48]. In our RNA sequencing dataset, SUV420H1 showed increased expression of the canonical isoform-coding transcript in lesional samples. While in non-lesional skin samples, a shorter isoform-coding transcript expression is elevated, missing the C-terminal region following the SET domain implicated in protein–protein interactions. While the isoform differentially expressed in non-lesional skin increases H4K20me3 levels globally in the nucleus, the canonical isoform-mediated methylation is mainly restricted to pericentric regions [161]. These alterations may impact psoriasis since SUV4-20 members are crucial for DNA replication [100], developmental DNA rearrangements [101], and telomeric chromatin formation [102].

##### PRDM Subfamily of Methyltransferases

The PRDMs are part of the SET domain family of histone methyltransferases, comprising 19 distinct transcription factors [162,163]. Although classified as methyltransferases, only some members exhibit this activity, including PRDM2 [53,54,164], MECOM (PRDM3) [55], PRDM7 [56], PRDM8 [57], PRDM9 [58,59,60,61,62], and PRDM16 [55,63]. These proteins are critical in regulating cell proliferation, differentiation, and gene expression through various signaling pathways [165,166]. In lesional skin, altered expression levels of PRDM2, MECOM, and PRDM8 were observed. PRDM2 is vital for stem cell self-renewal and cellular quiescence [108], and it regulates T cell-specific transcription factor GATA3 activity [109], whose levels are reduced in psoriatic lesional skin compared to non-lesional samples. Tape-stripping non-lesional areas also decreases GATA3 expression, indicating its role in inflammation and epidermal regeneration [167]. Interestingly, the increased expression of a PRDM2 transcript variant coding a 44-amino-acid-sized micropeptide with unknown regulatory function was detected in lesions. MECOM’s altered expression in psoriasis was previously shown to correlate with excessive keratinocyte proliferation [105]. In addition, MECOM is essential for hematopoiesis [168], inhibiting monocyte differentiation into macrophages [106]. However, the transcript variant of MECOM with elevated expression in lesions codes for an inactive isoform where the PR domain is missing, based on our database. Such isoforms typically act as inhibitors in a competitive manner. Therefore, further studies are required to elucidate the precise function of MECOM. PRDM8 is a key player in inducing trained immunity in response to damage-associated molecular patterns, relevant to chronic inflammatory diseases [169]. We found decreased expression of the canonical PRDM8 isoform-coding transcript.

##### Other SET Domain-Containing Histone Lysine Methyltransferases: SETD7 and SETD8

SETD7 is expressed normally in non-lesional and lesional psoriatic skin, while SETD8 shows altered expression in lesions. In particular, the expression of a shorter isoform-coding transcript is elevated, containing a reduced-size SET domain with unknown activity. SETD8 specifically catalyzes H4K20me1 methylation [50], while SUV4-20H1/H2 (discussed above) converts H4K20me1 to H4K20me2/3, which is essential for pre-replication complex formation and cell cycle progression [170]. SETD8 also supports the survival and differentiation of epidermal stem cells [104], suggesting that its altered expression may contribute to psoriatic changes in cell proliferation and differentiation.

Further discussion on SET domain methyltransferases with no or uncertain histone methyltransferase activity is provided as Appendix A [171,172,173,174,175,176,177,178,179,180].

#### 3.1.2. Seven-β-Strand (7BS) Methyltransferases

The seven-β-strand methyltransferases are primarily known as non-histone-specific methyltransferases; nevertheless, several members may also function as lysine or arginine histone methyltransferases [68,69,181,182]. Therefore, further discussion on seven-β-strand methyltransferases is provided as Appendix A [67,69,182,183,184,185,186,187,188,189,190,191,192,193,194,195,196,197,198,199,200,201,202,203,204,205,206,207,208,209,210,211,212,213,214,215,216,217].

## 4. Materials and Methods

### 4.1. Guidelines for Establishing a Combined Psoriasis Transcriptome Sequencing Dataset Based on Literature Sources

Our comprehensive transcriptome sequencing database was assembled as described previously [9,23]. In brief, data from three psoriatic transcriptome studies [20,21,22] of randomly recruited patients with chronic plaque psoriasis and healthy individuals were merged. In these studies, RNA sequencing data were obtained from 6 mm skin punch biopsies collected from various regions of the body, without any age (>18) or gender criteria (non-lesional psoriatic: *n* = 27, lesional psoriatic: *n* = 99, and healthy individuals: *n* = 172). PASI constitutes at least 1% of the total body surface area. To ensure that our data accurately reflect the general alterations associated with chronic plaque psoriasis, defined washout periods were implemented before biopsy collection for both topical (1 week) and systemic (2 weeks) treatments.

### 4.2. Processing and Differential Expression Analysis of RNA Sequencing Data

Data processing was performed as described previously [9,23]. Differential expression analysis was performed on the previously published dataset [23]. In brief, the RNA sequencing data were sourced from the Sequence Read Archive under accession numbers SRP035988, SRP050971, and SRP055813 utilizing SRA-tools (version 2.9.2). All samples of the datasets were uniformly reprocessed to ensure consistent analysis. Transcript levels were assessed employing Kallisto [218] (version 0.43.0) and the GENCODE [219] v27 transcriptome annotation, with Kallisto (defined parameters: --bias --single -l 120 -s 20 -b 100). Subsequently, transcript-level length-scaled TPM (Transcripts Per Million) expression estimates computed by Kallisto were transferred into the R statistical environment (version 3.4.3.) using the tximport [220] package (version 1.6.0). Following TMM normalization (using edgeR [221] v3.20.9) and voom transformation (limma [222,223] v3.34.9), the voomWithQualityWeights() function was employed, integrating sample-specific weights with transcript-level weights to accommodate lower-quality samples while mitigating their influence. Expression differences across sample groups were assessed using Limma. A linear model was applied via the lmFit function, and moderated t-statistics were computed using eBayes. Transcripts with an FDR-corrected [222,224] *p*-value of <0.05 were considered as differentially expressed.

### 4.3. Screening for Histone Methylation-Related DETs in Psoriasis

Datasets downloaded from https://amigo.geneontology.org/amigo (accessed on 24 April 2024) were employed to analyze differentially expressed transcripts (DETs) from the non-lesional/uninvolved (NL) vs. healthy (H) and lesional (L) vs. healthy (H) comparisons. The downloaded methyltransferase dataset (GO:0042054 and Appendix A) was augmented and verified with relevant information extracted from the literature that also includes methyltransferase-specific complexes. The literature references used are listed in Table 1. Detailed information about the dataset used for the screening is presented in Figure 1a–c; where data from both the GO database and the literature [26,27,28,29,30,31,32,33,34,35,36,37,38,39,40,41,42,43,44,45,46,47,48,49,50,51,52,53,54,55,56,57,58,59,60,61,62,63,67,69,225,226,227] are presented. Intersection analysis was used to filter and identify matches between non-lesional and healthy, as well as lesional and healthy samples, and the downloaded methyltransferase datasets in Python (Python 3.13.0).

### 4.4. Analysis of Protein Isoforms Derived from Differentially Expressed Transcripts

Differentially expressed transcripts in our psoriasis database were matched with Transcript IDs from the Ensembl database (https://www.ensembl.org/ accessed on 20 May 2025). Protein-coding transcripts were assigned their corresponding UniProt identifiers. In our analysis, the “canonical” protein isoform listed in UniProt served as the reference for comparing alternative isoforms. In our comparative analysis of DET-encoded protein isoforms, we considered amino acid deletions and sequence variations documented in the UniProt database for all alternative isoforms where these differences were explicitly noted. For isoforms lacking detailed sequence information in UniProt, we performed sequence comparisons against the canonical sequence using the Protein BLAST tool (https://blast.ncbi.nlm.nih.gov accessed on 21 May 2025), with particular focus on identifying gaps and mismatches.

Using the protein-coding differentially expressed transcripts (DETs), the domains of various isoforms were identified based on UniProt protein sequences, utilizing the Pfam databases of InterPro (https://www.ebi.ac.uk/interpro/ accessed on 22 May 2025). To ensure accurate proportional representation in the figures, the Prosite MyDomains Image Creator tool (https://prosite.expasy.org/ accessed on 24 May 2025) was employed for visualization.

## 5. Conclusions

In summary, various subfamilies of histone methyltransferases, including the EZ, SET1, SET2, SMYD, SUV39, SUV4-20, and PRDM subfamilies, play crucial roles in regulating gene expression, cell proliferation, and differentiation. Dysregulation of key proteins, such as SETD8 and EZH2, may contribute to hyperproliferation of keratinocytes and inflammatory responses related to the disease. Abnormal expressions of proteins like MECOM and PRDM2 further indicate their significance in immune modulation and stem cell functions, likely to influence the pathogenesis of psoriasis. Our analysis not only confirmed previously reported expressional alteration of EHMT1/2 in non-lesional skin but also revealed abnormal transcription of two SET domain family members and a β7 histone MTase in non-lesional skin. Transcriptional changes of these MTases highlight their potential involvement in early dysregulation of keratinocyte and T cell proliferation and differentiation. Additionally, the interactions of these methyltransferases with other signaling pathways highlight their potential as therapeutic targets for managing psoriasis symptoms. It is important to note that since our study is based on mRNA expression data analysis, further research is required to determine how these transcriptional changes manifest at the protein level and whether they affect enzymatic activity, function, and downstream cellular processes. However, if translated, the differentially expressed transcripts identified in this study may give rise to histone methyltransferase isoforms that influence the modification of key histone lysine residues, including H3K27 (EZH2, EZH1), H3K36 (ASH1L, NSD1, NSD2, SETD2), H3K9 (EHMT1, EHMT2, SUV39H2, SETDB2), and H4K20 (SUV420H1, SETD8). Understanding these complex regulatory mechanisms will be essential for developing new strategies to treat and combat psoriasis effectively.

## Figures and Tables

**Figure 1 ijms-26-06329-f001:**
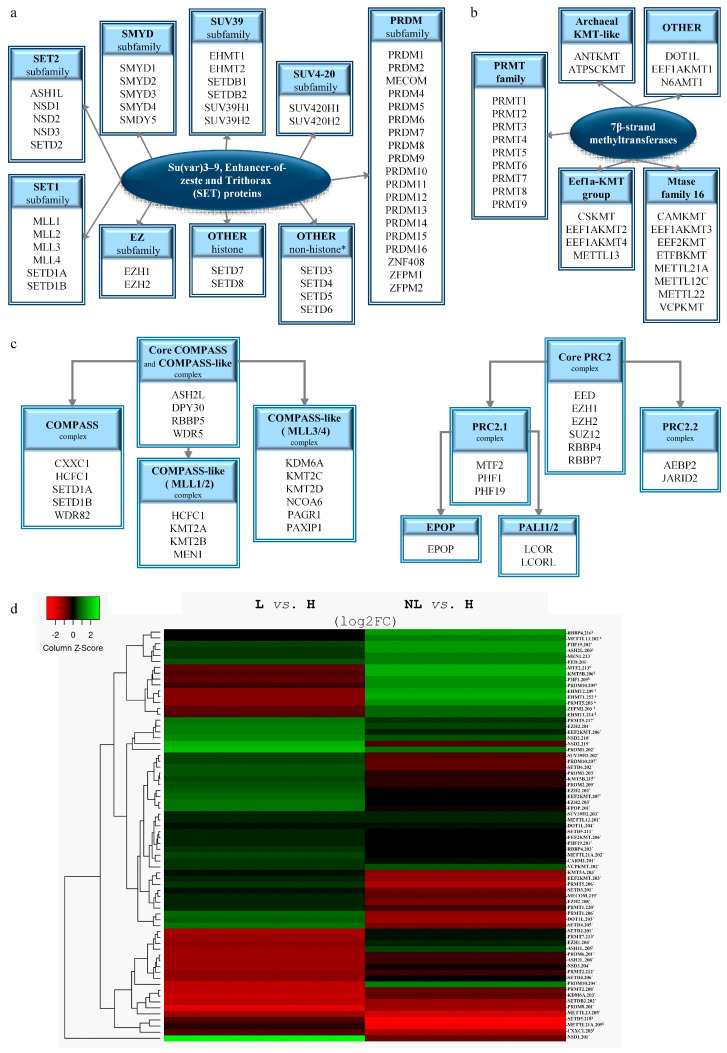
Differentially expressed histone methylation-related molecules in psoriasis. Classification of SET domain (**a**) and 7β-strand (**b**) lysine and/or arginine methyltransferases and their complexes (COMPASS, COMPASS-like, and PRC2) required for their proper function (**c**) used for screening. The heatmap of differentially expressed transcripts of methyltransferases and associated complex members in psoriasis (**d**). (H: healthy, NL: non-lesional, L: lesional skin; ¥: transcript variants showing differential expression in NL vs. H; ‡: indicate transcript variants that are differentially expressed in both NL and L skin vs. to H; *: transcript variants showing disparate expression levels in L skin vs. H).

**Figure 2 ijms-26-06329-f002:**
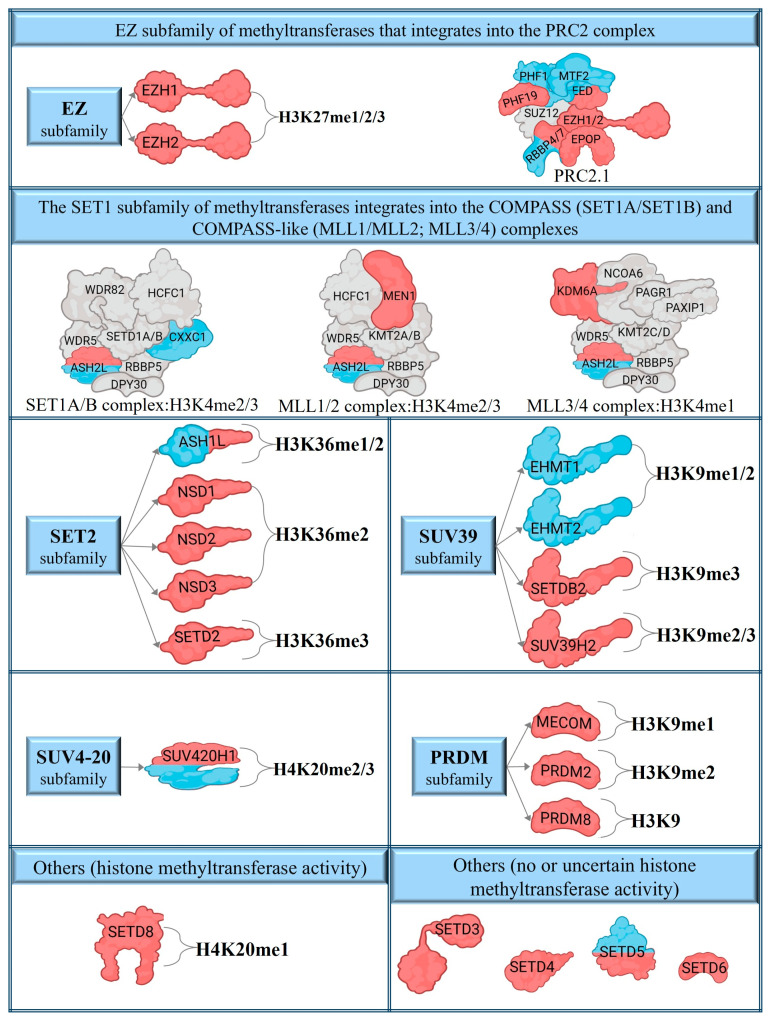
The differentially expressed transcripts of SET domain catalytic methyltransferases and their complexes in psoriasis. The blue colors indicate differentially expressed transcripts in non-lesional skin, while red indicates lesional transcriptional alterations. Blue and red colors indicate the disparity in both non-lesional and lesional expression levels.

**Figure 3 ijms-26-06329-f003:**
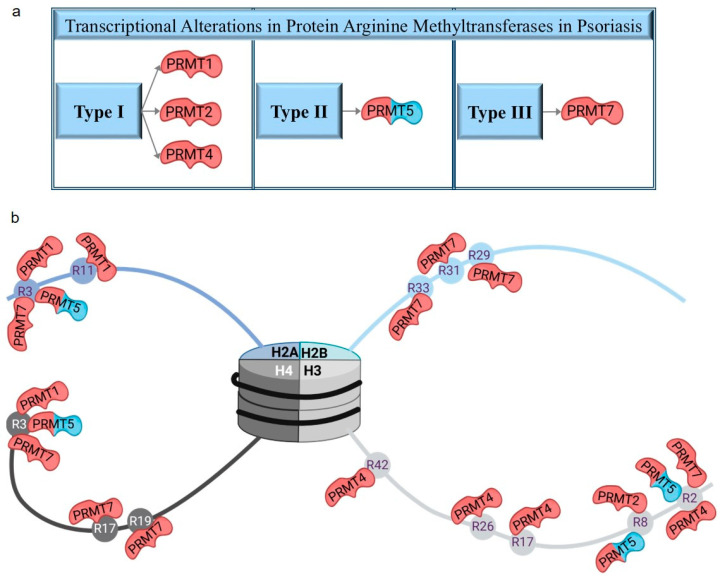
Protein arginine methyltransferases with altered expression in psoriasis (**a**) and their target histones with arginine methylation sites (**b**). The red colors highlight differentially expressed transcripts in lesions, while the blue and red colors reflect differences in expression levels in both lesional and non-lesional skin areas compared to healthy controls.

**Figure 4 ijms-26-06329-f004:**
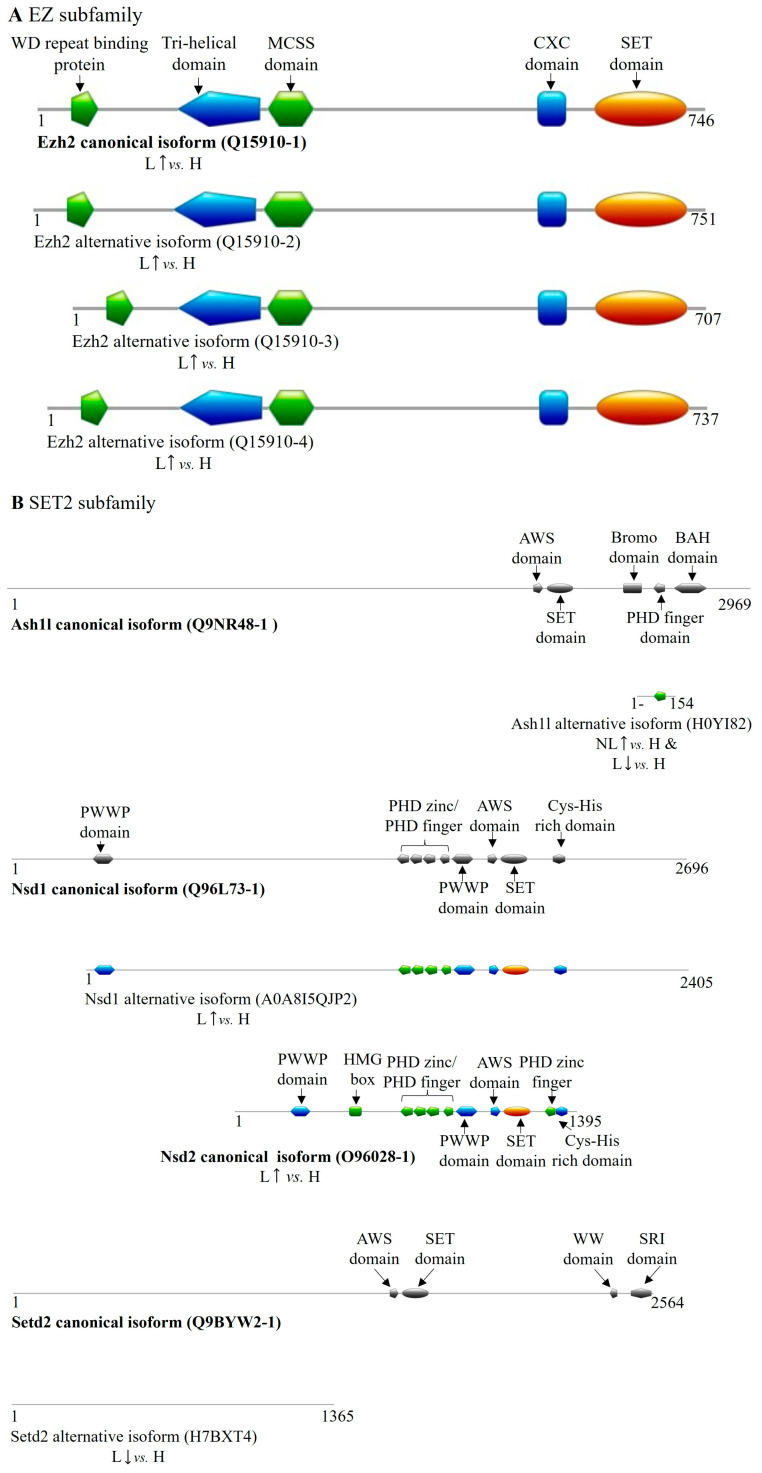
Differentially expressed transcript variant-encoded protein isoforms of SET domain family methyltransferases in psoriasis. DET-encoded isoforms of (**A**) EZ, (**B**) SET2, (**C**) SUV39, (**D**) SUV4-20, (**E**) other SET domain-containing methyltransferases; the (**F**,**G**) PRDM (PR/SET domain) subfamily members are illustrated in color. Canonical isoforms, if expressed normally, are depicted in gray for isoform comparison. (H: healthy, NL: non-lesional, L: lesional, ↑: increased expression of coding transcript, ↓: decreased expression of coding transcript).

**Table 1 ijms-26-06329-t001:** SET domain-containing histone methyltransferases and their constituents that target histone lysine residues for methylation. Methyltransferases identified with differentially expressed transcripts in non-lesional (NL) and/or lesional (L) skin are highlighted in bold.

	Transcriptional Alterations in SET Domain Lysine Methyltransferases in Psoriasis				
Subfamily	Gene ID	Alternative Name	Histone Modification	Reference	Differentially Expressed
**EZ subfamily**	**EZH1**	KMT6B	H3K27me1/2/3	[26,27]	**L vs. H**
	**EZH2**	KMT6A			**L vs. H**
**SET1 subfamily**	MLL1	KMT2A	H3K4me2/3	[28]	-
	MLL2	KMT2B			-
	MLL3	KMT2C	H3K4me1		-
	MLL4	KMT2D			-
	SETD1A	KMT2E	H3K4me2/3		-
	SETD1B	KMT2F			-
**SET2 subfamily**	**ASH1L**	KMT2H	H3K36me1/2	[29]	**NL vs. H; L vs. H**
	**NSD1**	KMT3B	H3K36me2	[30]	**L vs. H**
	**NSD2**	KMT3G		[30,31]	**L vs. H**
	**NSD3**	KMT3F		[30]	**L vs. H**
	**SETD2**	KMT3A	H3K36me3	[32]	**L vs. H**
**SMYD subfamily**	SMYD1	KMT3D	Unknown	Unknown	-
	SMYD2	KMT3C	H3K36me2	[33]	-
	SMYD3	KMT3E	H4K5; H4K20me3	[34,35]	-
	SMYD4	-	Unknown	Unknown	-
	SMYD5	-	H4K20me3; H3K36me1; H3K37me1	[36,37]	-
**SUV39H subfamily**	**EHMT1**	KMT1D	H3K9me1/2	[38]	**NL vs. H**
	**EHMT2**	KMT1C		[38,39]	**NL vs. H**
	SETDB1	KMT1E	H3K9me1/2/3	[40,41,42]	-
	**SETDB2**	KMT1F	H3K9me3	[43]	**L vs. H**
	SUV39H1	KMT1A	H3K9me2/3	[44,45,46]	-
	**SUV39H2**	KMT1B			**L vs. H**
**SUV4-20 subfamily**	**SUV420H1**	KMT5B	H4K20me2/3	[47,48]	**NL vs. H; L vs. H**
	SUV420H2	KMT5C			-
**Others**	SETD7	KMT7	H3K4me1	[49]	-
	**SETD8**	KMT5A	H4K20me1	[50,51]	**L vs. H**
**RIZ (PRDM) subfamily (PR/SET domain)**	**PRDM1**	BLIMP1	pseudo-MTase	[52]	**L vs. H**
	**PRDM2**	KMT8A	H3K9me2	[52,53,54]	**L vs. H**
	**MECOM**	KMT8E	Unclear (H3K9me1)	[52,55]	**L vs. H**
	PRDM4	PFM1	Unknown	Unknown	-
	PRDM5	PFM2	pseudo-Mtase	[52]	-
	**PRDM6**	KMT8C			**L vs. H**
	PRDM7	ZNF910	H3K4me3	[52,56]	-
	**PRDM8**	KMT8D	Unclear (H3K9)	[52,57]	**L vs. H**
	PRDM9	KMT8B	H3K4me1/2/3; H3K9me1/3; H3K18me1; H3K36me3; H4K20me1/2	[52,58,59,60,61,62]	-
	**PRDM10**	PFM7	pseudo-MTase	[52]	**NL vs. H; L vs. H**
	PRDM11	PFM8	Unknown	Unknown	-
	PRDM12	PFM9	pseudo-MTase	[52]	-
	PRDM13	PFM10	Unknown	Unknown	-
	PRDM14	PFM11	pseudo-MTase	[52]	-
	PRDM15	ZNF297			-
	PRDM16	KMT8F	Unclear (H3K9me1; H3K4me3)	[52,55,63]	-
	ZNF408	PRDM17	Unknown	Unknown	-
	ZFPM1	FOG1			-
	**ZFPM2**	FOG2			**NL vs. H**

**Table 2 ijms-26-06329-t002:** The molecular compositions of histone methyltransferase COMPASS/COMPASS-like and PRC2 complexes. (Transcripts exhibiting altered expression in psoriasis are shown in bold).

MTase Complex	Subtypes of Complex		Gene ID	Alternative Name	Differentially Expressed
PRC2 complex (EZ subfamily)	Core components of PRC2 complex		**EED**	HEED	**NL vs. H; L vs. H**
			**EZH1**	KMT6B	**L vs. H**
			**EZH2**	KMT6A	**L vs. H**
			SUZ12	JJAZ1	-
			**RBBP4**	RbAp48	**NL vs. H; L vs. H**
			RBBP7	RbAp46	-
	PRC2.1	PRC2.1	**MTF2**	PCL2	**NL vs. H**
			**PHF1**	PCL1	**NL vs. H**
			**PHF19**	PCL3	**L vs. H**
		EPOP-PRC2.1	**EPOP**	C17orf96	**L vs. H**
		PALI1/2-PRC2.1	LCOR	C10orf12	-
			LCORL	MLR1	-
	PRC2.2	AEBP2-PRC2.2	AEBP2	-	-
		JARID2-PRC2.2	JARID2	JMJ	-
COMPASS- and COMPASS-like complex (SET1 subfamily)	Core components of COMPASS and COMPASS-like complex		**ASH2L**	ASH2L1	**NL vs. H; L vs. H**
			DPY30	HDPY-30	-
			RBBP5	SWD1	-
			WDR5	SWD3	-
	COMPASS complex		**CXXC1**	PHF18	**NL vs. H**
			HCFC1	HFC1	-
			SETD1A	KMT2F	-
			SETD1B	KMT2G	-
			WDR82	TMEM113	-
	COMPASS-like complex (MLL1/2)		HCFC1	HFC1	-
			KMT2A	MLL1	-
			KMT2B	MLL2	-
			**MEN1**	MENIN	**L vs. H**
	COMPASS-like complex (MLL3/4)		**KDM6A**	UTX	**L vs. H**
			KMT2C	MLL3	-
			KMT2D	MLL4	-
			NCOA6	RAP250	-
			PAGR1	C16orf53	-
			PAXIP1	PAXIP1L	-

**Table 3 ijms-26-06329-t003:** Lysine methyltransferases characterized by seven-β-strand structures, together with their targets on histone or non-histone proteins and the type of modifications [67,68]. Methyltransferases identified with differentially expressed transcripts in non-lesional (NL) and/or lesional (L) skin are highlighted in bold.

Transcriptional Alterations in 7β-Strand Lysine Methyltransferases in Psoriasis						
Group	Gene ID	Alternative Name	Substrate	Modification	Signaling Regulation	Differentially Expressed
**Archaeal KMT-like**	ANTKMT	FAM173A	ANT1/2	K52me3	Mitochondrial metabolism	-
	ATPSCKMT	FAM173B	ATP synthase c-subunit	K43me3		-
**Eef1a-KMT group**	**CSKMT**	METTL12	Citrate synthase	K368me1/2/3 or K395me1/2/3	Mitochondrial metabolism	**L vs. H**
	EEF1AKMT2	METTL10	eEF1A	K318me3	mRNA translation	-
	EEF1AKMT4	ECE2	eEF1A	K36me2/3		-
	**METTL13**	EEF1AKNMT	eEF1A	K55me2		**NL vs. H**
**Mtase family 16**	CAMKMT	C2orf34	Calmodulin	K115me3	Neural development	-
	EEF1AKMT3	METTL21B	eEF1A	K165me1/2/3	mRNA translation	-
	**EEF2KMT**	FAM86A	eEF2	K525me3		**L vs. H**
	ETFBKMT	METTL20	ETFβ	K200me2/3; K203me2/3	Mitochondrial metabolism	-
	**METTL21A**	FAM119A	HSPA1; HSPA5; HSPA8	K561me3; K585me3; K565me3	Chaperones/protein stability	**NL vs. H;** **L vs. H**
	METTL21C	C13orf39	HSPA8; VCP/p97	K561me3; K315me3		-
	METTL22	C16orf68	KIN17	K135me3	Chromatin regulation	-
	**VCPKMT**	METTL21D	VCP/p97	K315me3	Chaperones/protein stability	**L vs. H**
**Others**	**DOT1L**	KMT4	Histone H3	K79me1/2/3	Chromatin regulation	**L vs. H**
	EEF1AKMT1	N6AMT2	eEF1A	K79me3	mRNA translation	-
	N6AMT1	KMT9	Histone H4	K12me1	Chromatin regulation	-

**Table 4 ijms-26-06329-t004:** The classification of PRMT histone methyltransferases and their targets on histones and the type of modifications [69]. Methyltransferases identified with differentially expressed transcripts in non-lesional (NL) and/or lesional (L) skin are highlighted in bold.

Types of PRMTs	Gene ID	Alternative Name	Histone Modification	Differentially Expressed
**Type I.**	**PRMT1**	HRMT1L2	H2AR3me2; H2AR11me2; H4R3me2	**L vs. H**
	**PRMT2**	HRMT1L1	H3R8me2	**L vs. H**
	PRMT3	HRMT1L3	H4R3me2	-
	**PRMT4**	CARM1	H3R2me2; H3R17me2; H3R26me2; H3R42me2	**L vs. H**
	PRMT6	HRMT1L6	H2AR3me2; H2AR11me2; H2AR29me2; H3R2me2; H3R42me2; H4R3me2	-
	PRMT8	HRMT1L3	H4R3me2	-
**Type II.**	**PRMT5**	HRMT1L5	H2AR3me1/2; H3R2me1/2; H3R8me2; H4R3me2	**NL vs. H; L vs. H**
	PRMT9	PRMT10	-	-
**Type III.**	**PRMT7**	KIAA1933	H2AR3me1; H2BR29me1; H2BR31me1; H2BR33me1; H3R2me1/2; H4R3me1; H4R17me1; H4R19me1	**L vs. H**

**Table 5 ijms-26-06329-t005:** Differential expression of transcripts from the SET domain methyltransferase family observed in psoriasis, and the protein isoforms they encode.

Subfamily	Gene ID	Transcript ID	Transcript Type	log2fc L vs. H	FDR L vs. H	log2fc NL vs. H	FDR NL vs. H	Uniprot Protein ID
EZ subfamily	EZH1	ENST00000585550.5	Retained intron	−1.165	4.17 × 10^−2^	0.606	5.39 × 10^−1^	-
	EZH2	ENST00000320356.6	Protein-coding	2.625	1.73 × 10^−5^	0.773	5.94 × 10^−1^	Q15910-2
		ENST00000460911.5		2.342	6.71 × 10^−11^	0.166	8.70 × 10^−1^	Q15910-1
		ENST00000350995.6		2.050	1.58 × 10^−4^	0.172	9.12 × 10^−1^	Q15910-3
		ENST00000483967.5		1.239	9.00 × 10^−3^	−0.627	5.79 × 10^−1^	Q15910-4
SET2 subfamily	ASH1L	ENST00000492987.2	Nonsense-mediated decay	−1.342	8.29 × 10^−6^	0.837	4.52 × 10^−3^	H0YI82
	NSD1	ENST00000347982.8	Protein-coding	5.601	1.18 × 10^−6^	−0.485	9.06 × 10^−1^	A0A8I5QJP2
	NSD2	ENST00000508803.5	Protein-coding	3.598	2.25 × 10^−9^	−0.222	9.09 × 10^−1^	O96028-1
		ENST00000482415.6	Processed transcript	2.818	1.91 × 10^−6^	0.902	4.90 × 10^−1^	
	NSD3	ENST00000525081.1	Processed transcript	−1.255	3.37 × 10^−9^	0.046	9.18 × 10^−1^	-
	SETD2	ENST00000330022.11	Nonsense-mediated decay	−1.442	6.93 × 10^−3^	0.424	6.10 × 10^−1^	H7BXT4
SUV39 subfamily	EHMT1	ENST00000640639.1	Protein-coding	−0.816	2.68 × 10^−1^	1.989	1.81 × 10^−2^	A0A1W2PPZ7
	EHMT1	ENST00000488242.2	Processed transcript	−0.158	6.96 × 10^−1^	1.108	9.57 × 10^−3^	-
	EHMT2	ENST00000477678.1	Retained intron	−0.855	8.12 × 10^−2^	1.906	5.07 × 10^−6^	-
	SETDB2	ENST00000317257.12	Protein-coding	−1.932	4.00 × 10^−2^	−0.997	5.13 × 10^−1^	Q96T68-2
	SUV39H2	ENST00000354919.10	Protein-coding	1.715	1.89 × 10^−7^	−0.304	7.42 × 10^−1^	Q9H5I1-1
	SUV39H2	ENST00000358298.6		1.236	3.74 × 10^−2^	0.594	6.19 × 10^−1^	H0Y306
SUV4-20 subfamily	SUV420H1 (KMT5B)	ENST00000615954.4	Protein-coding	1.715	2.63 × 10^−2^	−0.027	9.92 × 10^−1^	Q4FZB7-1
		ENST00000405515.5		−0.080	9.30 × 10^−1^	1.985	6.14 × 10^−3^	Q4FZB7-2
Others	SETD8 (KMT5A)	ENST00000437502.1	Protein-coding	1.025	1.81 × 10^−2^	−0.914	3.44 × 10^−1^	C9JKQ0
RIZ (PRDM) subfamily (PR/SET domain)	PRDM1	ENST00000369091.6	Protein-coding	3.995	1.94 × 10^−4^	1.265	5.76 × 10^−1^	O75626-2
		ENST00000369096.8		1.745	9.12 × 10^−40^	−0.023	9.34 × 10^−1^	O75626-1
	PRDM2	ENST00000491134.5	Nonsense-mediated decay	1.778	5.48 × 10^−5^	−0.080	9.51 × 10^−1^	H0Y9J3
	MECOM	ENST00000628990.2	Protein-coding	1.182	3.70 × 10^−3^	−0.375	7.08 × 10^−1^	Q03112-1
	PRDM6	ENST00000407847.4	Protein-coding	−1.334	4.97 × 10^−19^	−0.104	6.59 × 10^−1^	Q9NQX0-3
	PRDM8	ENST00000339711.8	Protein-coding	−2.606	1.18 × 10^−4^	−0.943	4.30 × 10^−1^	Q9NQV8-1
	PRDM10	ENST00000528746.5	Protein-coding	1.640	1.60 × 10^−2^	−0.342	8.43 × 10^−1^	E9PLV1
		ENST00000423662.6		−1.914	1.55 × 10^−2^	1.407	9.96 × 10^−2^	Q9NQV6-1
	ZFPM2	ENST00000517361.1	Protein-coding	−0.238	7.03 × 10^−1^	1.213	4.54 × 10^−2^	E7ET52

**Table 6 ijms-26-06329-t006:** The function of differentially expressed SET domain-containing histone methyltransferases and their known or potential roles in the pathogenesis of psoriasis.

Subfamily	Gene	Modulatory Role	Known Relevance to Psoriasis	Potential Relevance to Psoriasis
EZ subfamily	EZH2	Promotes keratinocyte proliferation and inflammation; downregulates miR-125a-5p, affecting TGFβ/SMAD pathway [15,76]	Drives keratinocyte hyperproliferation and inflammatory response in psoriatic skin [15,76]	-
SET2 subfamily	ASH1L	Modulates c-Myc activation and NF-κB signaling [77,78]	-	Affects keratinocyte proliferation/differentiation balance; suppresses TLR-induced TRAF6/NF-κB signaling modulating IL–17–mediated inflammation [79,80]
	NSD1	Modulates NF-κB via p65 methylation [81]; regulates proliferation via Wnt10b [82]		NF-κB is a key regulator of psoriatic inflammation [80]; altered expression of Wnt10b affects keratinocyte proliferation and cell migration [83]
	NSD2	Modulates Wnt/cyclin D1 pathway [84]		Elevated Cyclin D1 level contributes to hyperproliferation [85]
	SETD2	Regulates Th17/Treg via Lpcat4 [86], and AKT/mTOR signaling during wound healing [87]		Aberrant activation of mTORC1 signaling promoted hyperproliferation [88]; accelerated wound healing [89]; elevated Th17/Treg ratio [90]
SUV39 subfamily	EHMT1	Regulates cytokine expression via p50 [91]; modulates Treg function via FOXP3 [92]	-	NF-κB is a key regulator of psoriatic inflammation [80]; FOXP3-mediated Treg deficiency and excessive inflammation [93,94]
	SETDB2	Regulates mitosis [43]; IFN-I response in macrophages [95]		Regulation of keratinocyte proliferation, and M1/M2 macrophage imbalance [96]
	SUV39H2	Maintains basal keratinocyte stemness [97]		Keratinocyte proliferation, differentiation, and barrier formation and function [88,98,99]
SUV4-20 subfamily	SUV420H1	Controls DNA replication, telomere, and genome stability [100,101,102]	-	Keratinocyte proliferation and telomeric abnormalities [103]
Other SET domain-containing histone lysine methyltransferases	SETD8	Regulates cell proliferation and differentiation via p53, p63, and c-Myc [104]	-	Keratinocyte proliferation, differentiation, and barrier formation and function [88,98,99]
RIZ (PRDM) subfamily (PR/SET domain)	MECOM	Regulates cell proliferation [105]; monocyte/macrophage differentiation [106]	Altered MECOM expression correlates with increased keratinocyte proliferation in psoriatic lesions [105]	Increased tissue levels of TNFα⁺ monocytes/macrophages [107]
	PRDM2	Represses cell cycle genes (e.g., CCNA2) [108]; regulates Th function via GATA3 [109]	-	Increased CCNA2 [110] contribution to keratinocyte hyperproliferation; Th1/Th2 imbalance [111]

## Data Availability

Only publicly available data were used in the study (Sequence Read Archive, https://www.ncbi.nlm.nih.gov/sra (accessed on 15 November 2021); study ID: SRP035988, SRP050971, and SRP055813).

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
