# Peer review of "Unveiling the Role of Histone Methyltransferases in Psoriasis Pathogenesis: Insights from Transcriptomic Analysis"

_ijms, 2025, doi:10.3390/ijms26136329_

Round 1
Reviewer 1 Report
Comments and Suggestions for Authors
Psoriasis, a chronic immune-mediated skin disease characterized by keratinocyte hyperproliferation and immune dysregulation, is associated with molecular alterations present even in non-lesional skin. Epigenetic mechanisms, and in particular histone methylation, are thought to play a role in these pathological processes. Authors have engaged a bioinformatics-based transcriptomic profiling to systematically analyze the expression of histone lysine and arginine methyltransferases, including members of the SET-domain and seven-β-strand families. The analysis is based on a literature-curated transcriptome dataset that encompass nearly 300 samples of psoriatic and healthy skin. They also investigated the expression of components of associated methyltransferase complexes such as PRC2 and COMPASS. This analysis revealed widespread and distinct alterations in methyltransferase expression in both lesional and non-lesional psoriatic skin. Key molecules identified, like EZH2, SETD8, MECOM, and PRDM2, potentially contribute to keratinocyte proliferation, abnormal differentiation, and immune dysregulation. In addition, several methyltransferases present altered expression in non-lesional skin, suggesting their involvement in early disease development.
This comprehensive investigation can be considered as a foundational, systems-level overview of histone methylation-related transcriptional dysregulation in psoriasis that would propose novel epigenetic targets for therapeutic.
Major remarks
Conclusions are based on mRNA expression data analysis, which does not reflect post-transcriptional regulation, protein abundance, or enzymatic activity. This is particularly important for methyltransferases, whose function is highly context- and complex-dependent. These limitations must be more explicitly acknowledged
The transcriptomic dataset was constructed by merging samples from multiple studies. While this increases statistical power, it introduces potential heterogeneity in sample processing, sequencing platforms, or patient demographics. I therefore wonder how batch effect was taken into account. This biases introduced by the dataset integration is not clearly ackwoledge.
In several parts of the discussion mechanistic or causal roles for differentially expressed methyltransferases is inferred, like this one were changes in SETD8 or EZH2 expression are interpreted as drivers of keratinocyte proliferation, though no functional validation is provided. In my opinion these conclusions sounds more like « hypotheses » rather than definitive links since no functional validation is provided.
Why no GO term enrichment, pathway analysis or co-expression network has been done ? these analyses would contextualize all the gene expression changes to strenghten the biological interpretation.
Some genes show differential expression at the transcript isoform level, but the main discussion treats them as single-gene entities. Given the regulatory significance of alternative splicing in methyltransferases (e.g., PRDM family), this represents a lost opportunity. Considering isoform-level analysis would reveal functionally distinct regulation.
Minor typographic errors
Line 43 « extracellular matrix including modified splicing » add a coma after matrix,
Line 124 « CXXC1 shows expression changes in non-lesional skin » may be better to write « CXXC1 shows changes in expression in non-lesional skin..."
Line 129 « displays alterations » should be change in « displays alteration »
Line 327 « Alternations » Alterations
Author Response
Dear Reviewer,
We sincerely thank you for your kind comments and suggestions; your insights have been instrumental in supporting our work and enhancing the quality of the manuscript. We hope that you find the revised manuscript suitable for publication.
Major remarks
- Conclusions are based on mRNA expression data analysis, which does not reflect post-transcriptional regulation, protein abundance, or enzymatic activity. This is particularly important for methyltransferases, whose function is highly context- and complex-dependent. These limitations must be more explicitly acknowledged
According to your kind suggestion, the following text regarding the limitations of the study was added to the conclusion part of the manuscript:
It is important to note that since our study is based on mRNA expression data analysis, further research is required to determine how these transcriptional changes manifest at the protein level and whether they affect enzymatic activity, function, and downstream cellular processes.
- The transcriptomic dataset was constructed by merging samples from multiple studies. While this increases statistical power, it introduces potential heterogeneity in sample processing, sequencing platforms, or patient demographics. I therefore wonder how batch effect was taken into account. This biases introduced by the dataset integration is not clearly acknowledge.
To minimize batch effects and other biases, we applied several normalization and correction strategies. During the reprocessing phase, we employed normalization methods suitable for cross-study comparisons, such as TMM (trimmed mean of M-values) normalization using the edgeR package (version 3.20.9), to reduce non-biological variation. Additionally, we performed quality control measures and examined batch-related patterns in the data to ensure these did not confound our results. However, we recognize that residual batch effects could still influence the data. Therefore, in our data processing pipeline, transcript-level TPM expression estimates obtained from Kallisto were imported into the R environment (version 3.4.3) using the tximport package (version 1.6.0). To account for compositional differences and technical variability, we performed TMM normalization and subsequently applied the voom transformation with quality weights using limma’s voomWithQualityWeights() function. This approach allows for the integration of sample-specific and observation-level weights, downweighting lower-quality samples rather than excluding them entirely. Differential expression analysis was conducted with limma’s lmFit() function, and moderated t-statistics were derived using eBayes(). Transcripts were defined as differentially expressed if they met the FDR-adjusted p-value < 0 criteria. These steps were designed to robustly identify transcripts with significant expression differences while accounting for both technical and biological variability, including differences in sample quality.
The combination of normalization and correction strategies used in our analysis effectively minimizes the influence of batch effects and technical variability.
III. In several parts of the discussion mechanistic or causal roles for differentially expressed methyltransferases is inferred, like this one were changes in SETD8 or EZH2 expression are interpreted as drivers of keratinocyte proliferation, though no functional validation is provided. In my opinion these conclusions sounds more like « hypotheses » rather than definitive links since no functional validation is provided.
According to your kind suggestion, we have modified the text of the manuscript the following way:
In case no direct validation supporting the downstream role of methyltransferases was found in literature, conditional sentence mode (may, could) was applied to down-tone the sentences indicating their hypothetical nature (highlighted in yellow). While in those that are supported by direct evidence from literature were unchanged. To provide better clarity we added now a table (Table 6.) distinguishing between literature based known and suspected “Modulatory Role” of methyltransferases referring to them as, „Known Relevance to Psoriasis” and “Potential Relevance to Psoriasis”, respectively.
IV. Why no GO term enrichment, pathway analysis or co-expression network has been done? these analyses would contextualize all the gene expression changes to strengthen the biological interpretation.
The publications used to establish our database have previously conducted several analyses, including functional annotation using Gene Ontology, KEGG, Reactome, Biocarta, and Ingenuity Pathway Analysis (www.ingenuity.com), as well as co-expression network analysis. The results of these studies largely align with our compiled database (with differences only in sample size); therefore, we did not repeat these types of analyses to avoid redundancy.
The aim of our study was to provide a comprehensive overview of the potential involvement of elements participating in epigenetic regulation and to investigate their possible consequences. Our findings offer valuable insights for targeted future studies, which could either confirm or challenge the reported changes and their effects.
V. Some genes show differential expression at the transcript isoform level, but the main discussion treats them as single-gene entities. Given the regulatory significance of alternative splicing in methyltransferases (e.g., PRDM family), this represents a lost opportunity. Considering isoform-level analysis would reveal functionally distinct regulation.
According to your kind suggestion we performed an analysis of differentially expressed transcript encoded protein isoforms. In brief, we conducted a comprehensive analysis of differentially expressed transcript (DET)-derived protein isoforms in psoriasis, focusing on their structural and functional diversity. DETs were mapped to Ensembl and UniProt databases to identify corresponding protein-coding isoforms, using canonical sequences as reference. Where sequence data were incomplete, Protein BLAST was used to identify differences in amino acid composition. Special attention was given to methyltransferase families, where multiple DETs encoded alternative isoforms with domain alterations, truncations, or post-translational modification site losses. Notable examples include EZH2, ASH1L, SETD2, and PRDM family members, several of which encoded non-canonical or even inactive protein isoforms with potential regulatory roles. Translation of the differentially expressed transcripts may result in different isoforms of histone methyltransferases, potentially affecting the methylation status of essential histone lysine residues, including H3K27, H3K36, H3K9, and H4K20. These findings highlight the complexity and potential functional impact of isoform-level regulation in the pathogenesis of psoriasis, and we thank the reviewer for suggesting this analysis, which has substantially enriched the scope and interpretability of our results.
Details of the methodology and results are now provided in the newly added sections 4.4 (Analysis of Protein Isoforms Derived from Differentially Expressed Transcripts) and 2.5 (Diversity of Methyltransferase Transcript Variants and Encoded Isoforms in Psoriasis). The differential expression of SET-domain methyltransferase family transcripts and the corresponding protein isoforms identified in psoriasis are summarized in Table 5. Figure 4. illustrates representative differentially expressed transcript variants and their encoded isoforms. The abstract and discussion have been revised accordingly to reflect these new findings. Additionally, amino acid sequence information for the identified isoforms is now included in Supplementary Table 2. All modifications are highlighted in yellow within the manuscript.
Minor typographic errors
Line 43 « extracellular matrix including modified splicing » add a coma after matrix,
Line 124 « CXXC1 shows expression changes in non-lesional skin » may be better to write « CXXC1 shows changes in expression in non-lesional skin..."
Line 129 « displays alterations » should be change in « displays alteration »
Line 327 « Alternations » Alterations
Thank you for pointing out minor typographic errors which are now corrected in revised manuscript.
Once again, we sincerely thank you for reviewing our manuscript.
Reviewer 2 Report
Comments and Suggestions for Authors
This article offers an overview of histone methyltransferases and their expression patterns in psoriasis based on transcriptomic analysis datasets. While several review articles have explored the role of histones and their modifications in psoriasis, a focused analysis specifically on histone methyltransferases remains lacking. The data and tables presented in this article are clear and informative. My comments are as follows:
- Would it be possible to indicate the fold changes for each gene in the tables?
- Could you include a summary table for the Discussion section containing the gene symbol, known function, and relevance to psoriasis? Given that this article analyzes multiple genes, such a table would help readers quickly grasp the specific roles of these genes in psoriasis pathogenesis.
Author Response
Dear Reviewer,
We sincerely thank you for your kind comments and suggestions; your insights have been instrumental in supporting our work and enhancing the quality of the manuscript. We hope that you find the revised manuscript suitable for publication.
Comments and Suggestions for Authors
This article offers an overview of histone methyltransferases and their expression patterns in psoriasis based on transcriptomic analysis datasets. While several review articles have explored the role of histones and their modifications in psoriasis, a focused analysis specifically on histone methyltransferases remains lacking. The data and tables presented in this article are clear and informative. My comments are as follows:
Would it be possible to indicate the fold changes for each gene in the tables?
According to your kind suggestion we have provided the fold changes for each gene (as log2FC) not only in the supplementary table but also in Table 5.
Could you include a summary table for the Discussion section containing the gene symbol, known function, and relevance to psoriasis? Given that this article analyzes multiple genes, such a table would help readers quickly grasp the specific roles of these genes in psoriasis pathogenesis.
Following your kind suggestion, we have supplemented our discussion with the requested table highlighting literature based known and suspected “Modulatory Role” of methyltransferases referring to them as, „Known Relevance to Psoriasis” and “Potential Relevance to Psoriasis”, respectively (Table 6.).
Following the recommendations of the other reviewer, we conducted further analysis focusing on isoforms encoded by differentially expressed transcripts. As a result, the abstract has been revised, and new sections have been incorporated into the Results and Methods parts of the manuscript. The Discussion has also been substantially modified in light of the new data. All modifications are highlighted in yellow within the manuscript.
Round 2
Reviewer 1 Report
Comments and Suggestions for Authors
Thank you for the answer and modifications.